# Valorization of Sewage Sludge via Gasification and Transportation of Compressed Syngas

**Marek Mysior [1,\*]**, **Maciej Tomaszewski [2]**, **Paweł Stępień [2]**, **Jacek A. Koziel [3]** and **Andrzej Białowiec [2,3]**

[1] Department of Machine Design and Research, Wrocław University of Science and Technology, 7/9 Łukasiewicza Street, 50-371 Wroclaw, Poland

[2] Institute of Agricultural Engineering, Faculty of Life Sciences and Technology, Wroclaw University of Environmental and Life Sciences, 37/41 Chełmońskiego Street, 51-630 Wroclaw, Poland

[3] Department of Agricultural and Biosystems Engineering, Iowa State University, Ames, IA 50011-3270, USA

[\*] Correspondence: marek.mysior@pwr.edu.pl

**Abstract:** A significant challenge in the utilization of alternative gaseous fuels is to use their energy potential at the desired location, considering economic feasibility and sustainability. A potential solution is a compression, transportation in pressure tanks, and generation of electricity and heat directly at the recipient. In this research, the potential for generating syngas from abundant waste substrates was analyzed. The sewage sludge (SS) was used as an example of a bulky and abundant resource that could be valorized via gasification, compression, and transport to end-users in containers. A model was developed, and theoretical analyses were completed to examine the influence of the calorific value of the syngas produced from the SS gasification (under different temperatures and gasifying agents) on the efficiency of energy transportation of compressed syngas. First, the gasification simulation was carried out, assuming equilibrium in a downdraft gasifier (reactor) from 973–1473 K and five gasifying agents ($O_2$, $H_2$, $CO_2$, water vapor, and air). Molar ratios of the gasifying agents to the (SS) C ranged from 0.1–1.0. The model predicted syngas composition, lower calorific values (*LHV*) for a given molar ratio of the gasification agent, and compressibility factor. It was shown that the highest *LHV* was obtained at 0.1 molar ratio for all gasifier agents. The highest *LHV* (~20 MJ·(Nm$^3$)$^{-1}$) was obtained by gasification with $H_2$ and the lowest (~13 MJ·(Nm$^3$)$^{-1}$) in the case of air. Next, the available syngas volume in a compressed gas transportation unit and the stored energy was estimated. The largest syngas volume can be transported when $O_2$ is used as a gasifying agent, but the highest amount of transported energy was estimated for gasification with $H_2$. Finally, the techno-economic analyses showed that syngas from SS could be competitive when the energy of compressed syngas is compared with the demand of an average residential dwelling. The developed syngas energy transport system (SETS) concept proposes a new method to distribute compressed syngas in pressure tanks to end-users using all modes of transport carrying intermodal ISO containers. Future work should include the determination of energy demand for syngas compression, including pressure losses, heat losses, and analysis of the influence of syngas on storage and compression devices.

**Keywords:** sewage sludge; gasification; syngas compression; biorenewables; waste management; valorization; waste to carbon; waste to energy; circular economy; sustainability

## 1. Introduction

The energy sector in Europe is facing significant challenges due to increasing demand for electricity and heat, curbing greenhouse gas emissions and other efforts to reduce carbon footprint (e.g., waste

to energy, circular economy). According to the climate and energy framework for 2030 adopted by the European Council in 2014 and reviewed in 2018, one of the potential solutions is increased usage of renewable sources to 32.5% share in 2030. Biofuels are of great importance as to the gradual replacement of fossil fuels by renewables not only because of a need to reduce gaseous emissions but also because of the low diversity of fuels used worldwide. There are numerous research reports on the reduction of gaseous emissions in combustion through the utilization of biomass in energy generation [1,2] as well as attempts to increase the effectiveness of energy generation from fossil fuels, such as coal [3]. Sher et al. investigated the effect of the staging air injection location on the $NO_x$ and CO emissions and temperature profiles of a bubbling fluidized bed combustor using woody and non-woody biomass fuels [1]. It was shown, that a significant reduction of the $NO_x$ and CO emissions is possible by changing the process parameters. In a different work [2], authors investigated the effects of the combustion atmosphere and oxygen concentration in the oxidant of the oxy-fuel combustion of woody and non-woody biomass on gas emissions. Obtained results confirmed that a significant reduction in the $NO_x$ and CO emissions is possible by changing the combustion atmosphere. Zhang et al. investigated varying structures of slotting, gas velocities from nozzles and jet inclination angles on the $NO_x$ emissions and corrosion-subjected area [3]. Results of this work promote efforts to reduce the corrosive area in pulverized-coal furnaces or boilers, optimizing energy generation and gaseous emissions from coal combustion.

Despite successful efforts in gaseous emissions using fossil fuels and various forms of biomass, there is still a limited diversity of fuels used, which negatively affects the energy sector worldwide. In this paper, authors concentrate on solving this problem by systematic efforts on the introduction of novel energy sources that can effectively substitute fossil fuels. This problem is clearly visible in Poland, where there is a growing demand for natural gas (NG), and it is expected to continue until 2030. The majority of the NG in Poland is imported (~78%), and a greater diversity of energy sources is needed. An introduction of gaseous biofuels distributed on a local level could address this need. One of the biggest challenges is still a very limited infrastructure used to transport gas in pipelines, especially in rural areas. One potential solution is to transport compressed gaseous fuels by road, rail, or water, thus increasing its local and regional availability.

The transport of compressed gaseous fuels, including compressed natural gas (CNG) is well known [4–10]. In recent years, new research has focused on the compression, transport, and use of other gaseous fuels as an alternative to NG such as biogas in both treated and untreated forms (crude biogas) [11–14] and syngas (a product of a gasification process). A significant challenge in the utilization of alternative gaseous fuels is to use their energy potential at the desired location, considering economic benefits and sustainability. The sale of electricity from the combustion of gas in the cogeneration unit (CHP, combined heat and power) to the grid is an inefficient process due to a large amount of heat generated, which is typically not utilized and wasted to the atmosphere. This apparent waste is an opportunity to improve the sustainability of gas fuel distribution and utilization systems [15] through compression, transportation, and generation of electricity and heat in the CHP units directly at the recipient [12].

The synthesis gas (so-called 'syngas') is a source that can be used to generate energy which can be used directly to supply stationary or mobile CHP units, enabling diversification of energy sources [16]. It can also be utilized by the chemical industry. Transport is required, preferably in compressed form, to supply syngas to recipients in remote and underserved areas [17]. Syngas could be produced from abundant waste and reduce its negative impact on the environment. Syngas is used in various industries as a byproduct of the CNG/LNG production [18] and vehicle fuel [19]. Syngas has lower environmental impact compared with the conventional NG combustion [20].

The usage of syngas as fuel in vehicles with ICE (internal combustion engine) was analyzed in [19]. Pradhan et al. described the influence of particle size and the moisture content of feedstock on the quality of syngas produced by gasification using air. It was reported that the most crucial factor that affects the efficiency of ICE powered by syngas is related to the low energy density of produced syngas.

It was shown, that increased $H_2$ content lowers the CO emissions, but at the same time increases the $NO_x$ emissions. However, there is lack of information on how the pressure under which syngas is fed to an ICE influences energy balance and specific emissions, which requires an additional study on the compressibility of syngas, which was not yet analyzed. Furthermore, the usage of syngas in transportation requires the storage and distribution of syngas, which was not yet described.

Additional concepts of utilizing syngas in energy production are being developed. Anghilante et al. [18] presented several power-to-SNG plant concepts for upgrading bio-syngas to LNG or CNG through steam electrolysis and catalytic methanation process. The sewage sludge (SS), wood and straw, was used in the process of obtaining LNG. Although it was shown, that a full thermal integrated process efficiency (close to the maximal theoretical efficiencies) were reached, there is a lack of information on the energy efficiency of gasification of SS alone. Gasifying SS (without it being mixed with other less predictable substrates) is more practical, similarly to obtaining compressed syngas, not LNG. Thus, for this to be possible, a study on the gasification process of SS is necessary as well as an analysis of compressed gas properties.

In this research, we focus on syngas, especially on the potential of generating syngas from abundant waste substrates such as the sewage sludge (SS). The SS is challenging to manage, bulky, and an abundant resource that could be valorized in the sustainable waste management system. In Poland, SS cannot be disposed of to landfills since 2016. As a result, most of the SS is used in agriculture or is subjected to thermal treatment [21]. Thus, there is a need to develop novel applications of syngas in the energy sector, which makes it possible to further diversify energy sources and to replace NG in some applications.

This research proposes a new method to utilize SS as an energy source that can be gasified and distributed to end-users in pressure vessels via all modes of transport capable of carrying intermodal ISO containers. This work presents a new transdisciplinary approach based on the synergy of managing SS (a low quality, well geographically dispersed waste that could be used as a solid fuel), SS gasification and syngas compression (high quality compressed gaseous fuel) to improve the logistics and economy of energy distribution. SS is used as a model of abundant biowaste for compressed syngas production and transport, and thus the concept presented here could be useful to inform waste-to-energy conversions in the circular economy.

*1.1. Sewage Sludge and Thermal Treatment*

The SS production rate continues to increase, which forces finding a more sustainable way of managing it to meet the requirements of both international (EU) and national (PL) laws [22]. SS is a watery waste susceptible to rotting, odors, and leaching. It contains significant amounts of heavy metals, toxins, xenobiotics, pharmaceuticals, and pathogenic organisms [23]. Therefore, more R&D to sustainably improve the SS management is needed.

The thermal treatment of SS has been proposed as a method of solving problems related to the local generation and own energy needs by wastewater treatment plants [24]. To date, a relatively small number of the thermal treatment of SS plants are in operation, and all of them are a high capacity type [25]. The SS gasification to produce compressed syngas would allow not only the management of SS in accordance with the developed waste hierarchy but also generate new opportunities to obtain by-products and products with high energy potential and value.

*1.2. Gasification*

Gasification is a process in which solid material containing large amounts of carbon in its internal structure (biomass, organic waste) is transformed into a gas form [26]. It is carried out at ~1023 K~1273 K, $0 < \lambda < 1$, aided by a gasification agent (e.g., air, steam, $O_2$, $CO_2$ or $H_2$) [27–31]. The gasifying agent concentration affects the reactions and the syngas composition. The gasification process has four stages [27]: Heating and drying, pyrolysis, gasification reactions between solid- and gas-phases, and gas-phase reactions. The main product (syngas) is made up of $H_2$, CO, $CH_4$, $CO_2$,

and $N_2$ (when the gasification medium is air). Syngas is most often used for the production of heat and electricity. However, it may be used by the chemical industry [23]. The ratio of the main and by-products (ash & tar) depends on the feedstock properties and process parameters [27].

### 1.3. Parameters Affecting Sewage Sludge Gasification

Research has shown, that the key parameters affecting the process are the selection and concentration of the gasifying agent, the composition of the feedstock and the process temperature. Depending on the gasifying agent, the higher heating value (*HHV*) of the syngas can range from 4~7 MJ·Nm$^{-3}$ for air and up to 40 MJ·Nm$^{-3}$ for $H_2$ [28]. Due to economic reasons, the most commonly used gasifying agent is air [27]. The second factor is the feedstock composition, especially, the C, H, and O content and ratios. The O/C and H/C ratios have a significant impact on the lower heating value (*LHV*). When the C content is low, the O/C and H/C ratio increases, and more non-flammables (e.g., $CO_2$ and $H_2O$) are generated. Therefore, high C content feedstocks (e.g., SS) are preferred [29]. The temperature affects the syngas composition and properties. The temperature controls the reaction rates and the formation of individual gas components. Moreover, higher temperature promotes self-cleaning of the syngas from tar [30].

### 1.4. Objectives

This research was aimed to build a parsimonious model for the SS gasification to optimize the process and to propose a novel concept for the distribution of compressed syngas to the end-users. The effects of the following were modeled:

- gasifying agents: $O_2$, $H_2$, $CO_2$, water vapor, and air,
- molar ratios between gasification agents and feedstock C, and
- temperature.

The obtained syngas results were analyzed for:

- percentage of main syngas components,
- the *LHV*, and
- options for compression and transport in pressurized tanks.

Finally, the developed model was used to estimate the theoretical energy content in a novel transport unit (a syngas energy transport system; SETS) that can be used to store and transport syngas to end-users. The proposed SETS concept and the research scope are presented in Figure 1. The SETS is novel because syngas is used as a mobile energy source. The SS gasification is followed by the syngas compression and transport to end-users using road, rail or water transport.

The novelty of this proposed transdisciplinary approach is due to the waste-to-energy concept, by application of a mobile syngas transport unit comprising of high-pressure vessels to distribute syngas (high-quality fuel) produced from SS (abundant waste; low-quality fuel). The mobile system concept makes it possible to utilize compressed syngas in remote areas, where no gasification process can be conducted. The mobile syngas transport unit is based on a standard ISO container with 68 connected vessels capable to store gas up to 20 MPa (200 bar). The advantage of the proposed system is based on an increased syngas volume and density in transport by compression. It is necessary to research the syngas compressibility and energy content for transport. The proposed SETS means that biowaste (SS) can be converted to fuel, increasing the syngas availability, storage and flexible delivery/use options.

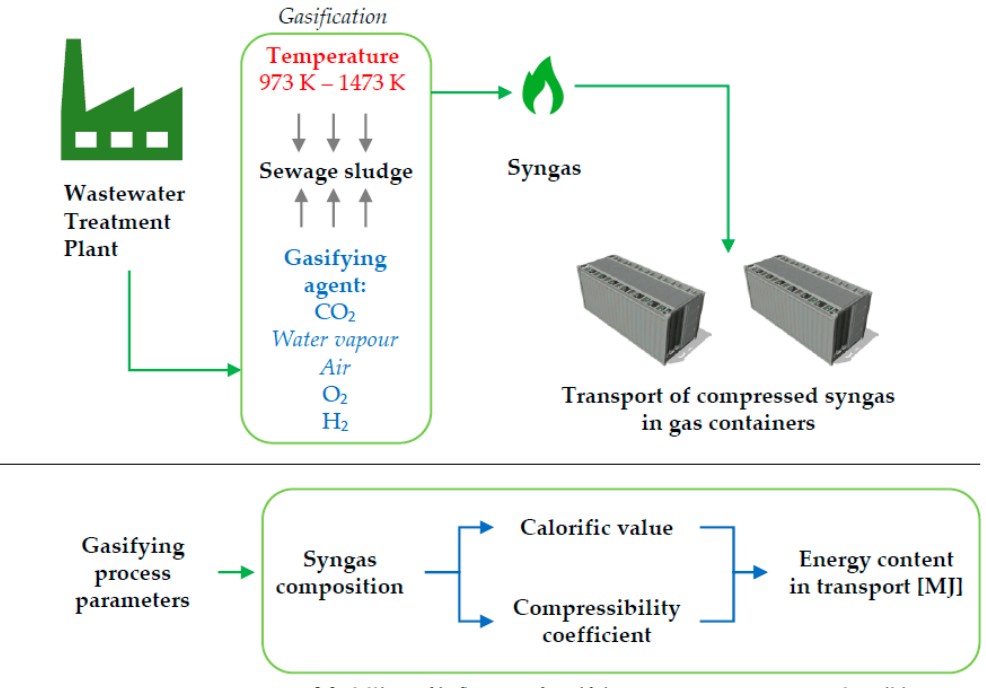

**Figure 1.** Optimization of sewage sludge gasification is part of a proposed syngas energy transport system (SETS) concept. The SETS is based on the waste-to-energy gasification of sewage sludge, compression of syngas, resulting in increased flexibility of storage, delivery and end-use options based on the standard ISO 20 feet containers equipped with smaller compressed gas tanks.

## 2. Materials and Methods

In this work, modeling of critical functionalities of the proposed SETS was performed. At first, theoretical modeling of the gasification process of SS was performed using the Gaseq Chemical Equilibrium Program 3.1 software. The novelty of the performed analysis is connected with the gasification of SS without any addition of other substrates such as wood or straw as well as with a range of performed analysis. The gasification process was analyzed for five different gasifying agents with variable molar numbers and 25 gasification temperatures. As a result, the chemical composition, and *LHV* of the syngas from SS were obtained. Results were compared with those available in the literature for verification of the proposed approach and validation of results.

Data obtained from the modeling of the SS gasification was used to develop and perform analytical modeling of the syngas compression to determine the energy content in a given volume of syngas as well as the compressibility coefficient that describes the state of obtained syngas at elevated pressure. For the scope of this work and the similarity of the application of the compressed syngas to CNG, the possibility of the application of models dedicated to determining the compressibility factor of NG was examined. Due to the obtained composition of syngas, there was a need to develop a model dedicated to the syngas based on several assumptions. This was done by the application of a real gas equation and method of weighting treatment to determine the syngas compressibility coefficient and syngas compression ratio describing the increase of volume under specific conditions.

As a result, the influence of the molar concentration of gasifying agents, and process temperature on the amount of energy possible to be stored and transported in the pressure vessel was obtained.

### 2.1. Characteristics of Sewage Sludge

The elemental composition and properties of SS were described elsewhere [28] (Table 1).

**Table 1.** Sewage sludge elemental composition and properties (based on [28]).

| Parameter | Value |
|---|---|
| C, % | 27.72 |
| H, % | 3.81 |
| N, % | 3.59 |
| O, % | 13.53 |
| S, % | 1.81 |
| Ash, % | 53.13 |
| Moisture, % | 5.30 |

### 2.2. Sewage Sludge Gasification Modeling Procedure

The following assumptions were made based on [32,33]. Specifically, the boundary conditions were established in those experimental studies where the gasification of SS was carried out. Therefore, the assumptions of the model were taken from experimental research and were as follows:

- gases are treated as semi-ideal,
- all elemental C from the SS is converted to gaseous components,
- process gases are limited to $CO$, $CO_2$, water vapor, $H_2$, $N_2$, $H_2S$, and $CH_4$,
- the total conversion of tar substances occurs in the oxidation zone,
- the temperatures of the solid- and gas-phases are the same,
- pressure losses are negligible (assumption of constant pressure).

The material was subjected to a thermal process at temperatures of 973, 998, 1023, 1048, 1073, 1098, 1123, 1148, 1173, 1198, 1223, 1248, 1273, 1298, 1323, 1348, 1373, 1398, 1423, 1448, and 1473 K. The equal incremental interval of 25 K was used. Such temperature selection is consistent with the model distribution of gasification temperatures, which occurs between 973 K and 1473 K [33]. The process pressure was set to 1013 hPa according to experimental conditions described in [33], and the authors wrote an assumption that the pressure is constant because there is a flow of gasifying agent and the resulting gas is withdrawn.

Five series of experiments were carried out for a different gasifying agent and its varying molar concentration. The following factors and their molar concentrations were used in the experiment:

- oxygen, from 0.1 to 1.0 moles (interval 0.1 mole),
- hydrogen, from 0.4 to 4.0 moles (interval 0.4 moles),
- carbon dioxide, from 0.2 to 2.0 moles (interval 0.2 mole),
- water vapor, from 0.2 to 2.0 moles (interval 0.2 m),
- air, from 0.1 to 1.0 moles (interval 0.1 mole) for oxygen and from 0.367 to 3.760 moles for nitrogen (interval 0.367 moles).

The oxygen demand required for the gasification process was calculated based on the chemical reactions taking place in the process summarized in Table 2. Table 3 presents the comparison of initial and boundary conditions of simulations.

The concentration of the gasifying agent results from the reactions described in Table 2. It presents the main reactions of the gasifying agent with the carbon element in equilibrium conditions. The number of moles in the equilibrium was divided into five. Additionally, the proportion of each factor was the same. Subsequently, values lower and higher than the equilibrium were determined.

**Table 2.** Gasification process reactions [32].

| Series | Reaction |
|---|---|
| Oxygen | $C + 0.5O_2 \rightarrow CO$ |
| Hydrogen | $C + 2.0H_2 \rightarrow CH_4$ |
| Carbon dioxide | $C + CO_2 \rightarrow 2.0CO$ |
| Water vapor | $C + H_2O \rightarrow CO + H_2$ |
| Air | $C + 0.5O_2 + 1.88N_2 \rightarrow CO + 1.88N_2$ |

**Table 3.** Data used to simulate the sewage sludge gasification.

| Parameter, Unit | | Value |
|---|---|---|
| The number of moles in the substrate | C | 2.3077 |
| | H | 3.7801 |
| | N | 0.2563 |
| | O | 0.8456 |
| | S | 0.0564 |
| | $H_2O$ | 0.2942 |
| The number of moles in the gasifying gas | O | 0.1 ÷ 1 (interval 0.1) |
| | H | 0.4 ÷ 4 (interval 0.4) |
| | $CO_2$ | 0.2 ÷ 2 (interval 0.2) |
| | Water vapor | 0.2 ÷ 2 (interval 0.2) |
| | Air | 0.1 ÷ 1 (interval 0.1) |
| The temperature of the gasification process, K | | 973 ÷ 1473 (interval 25) |
| Pressure, hPa | | 1013.25 |
| Initial concentration of gaseous products, % | CO $CO_2$ $H_2O$ (Water vapor) $H_2$ $N_2$ $CH_4$ | 0 |

The simulation of the SS gasification process was carried out using the Gaseq Chemical Equilibrium Program 3.1 software [34]. Based on the obtained synthesis gas composition, the *LHV* of 1 m³ of gas was calculated for normal conditions [35]:

$$LHV = \sum (V_S \cdot LHV_s) \tag{1}$$

where:

*Vs*—the percent component volume in the synthesis gas, %
*LHVs*—the theoretical calorific value of the synthesis gas component, MJ·m³ ($H_2$—10,748 MJ·m³, CO—12,634 MJ·m³, $CH_4$—35,725 MJ·m³, $H_2S$—23,152 MJ·m³).

### 2.3. Modeling of Gas Transport Volume of Syngas as a Function of Its Variable Composition

As a result of the SS gasification modeling, chemical compositions and *LHVs* were obtained for different gases as a result of the conducted analysis. Those values are required to determine the volume of gas in the transport and energy content of compressed syngas under specific storage conditions in a SETS. Those conditions were assumed based on the described syngas transport unit in the form of an ISO container with 68 pressure vessels, all connected into one storage volume of 14,960 m³. This volume is referred to as a 'water' volume of the storage unit and represents the amount of space gas can occupy when stored in the storage unit. The increase of gas volume is achieved by increased gas pressure in the compression process. This, in turn, makes it possible to store and transport more gas, which increases the energy content. Analytical calculations were carried out to determine the

gas volume in the transport unit. These calculations were based on the results of the SS gasification modeling and resulting chemical composition of syngas obtained for given gasification factors and the temperature of the gasification process. For this work, the real gas equation was used [36], which is based on the introduction of an additional coefficient to the ideal gas equation, the so-called 'gas compressibility coefficient' Z, which takes into account intermolecular interactions in the analyzed gas. In standard temperature and pressure (STP) conditions (273.15 K and absolute pressure of 100 kPa), the compressibility coefficient of real gases is assumed to be equal to one, which means gas under STP conditions may be treated as an ideal gas [36]. For most gases, with a pressure increase, the gas compressibility coefficient decreases, which means that real gas occupies a smaller volume as compared to an ideal gas at the same pressure and temperature [9]. The real gas equation constituting the basis for theoretical calculations is shown below (based on [36]):

$$V = \frac{Z \cdot n \cdot R \cdot T}{P} \tag{2}$$

where:

$P$—gas pressure, Pa,
$V$—gas volume, m$^3$,
$Z$—gas compressibility coefficient,
$n$—the number of moles of gas,
$R$—universal gas constant, J·mol$^{-1}$·k$^{-1}$,
$T$—gas temperature, K.

Equation (2) allows the determination of gas parameters at any temperature and volume, assuming the knowledge of gas compressibility coefficient. The $Z$ value can be determined using a series of equations and methods, depending on the composition of the analyzed gas. For natural gas, the most commonly used methods are GERG-2008, SGERG-88, and AGA-8 [37–39]. Authors have analyzed the possibility to apply methods mentioned above because there are many programs in which these three models are used for the determination of the natural gas compressibility factor. If the composition of syngas makes it suitable to use one of the above methods to determine the Z factor, it will make it possible to use already known solutions to determine the compressibility factor of syngas. Uncertainty of the compressibility factor using the GERG-2008 for single-phase mixtures is less than 0.1% in the normal range and 0.5% in the extended range of application [39]. The normal and extended range is related to the percentage of gas constituents, which represents applicability of this calculation method. However, due to the strictly defined scope of application, these methods cannot be directly applied to the syngas analysis due to the excessive amount of CO, $H_2O$, $H_2$, and $N_2$ in the obtained syngas. Figure 2 presents the list of modeled syngas compositions (minimum and maximum values) with the scope of application of the GERG-2008 method. This creates a need to determine a new approach in the determination of the syngas compressibility coefficient that will make it possible to calculate this value for the syngas for the obtained ranges of constituents (those are values obtained for the gasification of SS under conditions mentioned in Table 3 and are the result of this work):

- $CO_2$—up to 0.41 molar fraction,
- CO—up to 0.60 molar fraction,
- $H_2O$—up to 0.39 molar fraction,
- $H_2$—up to 0.78 molar fraction,
- $N_2$—up to 0.59 molar fraction,
- $H_2S$—up to 0.02 molar fraction,
- $CH_4$—up to 0.37 molar fraction.

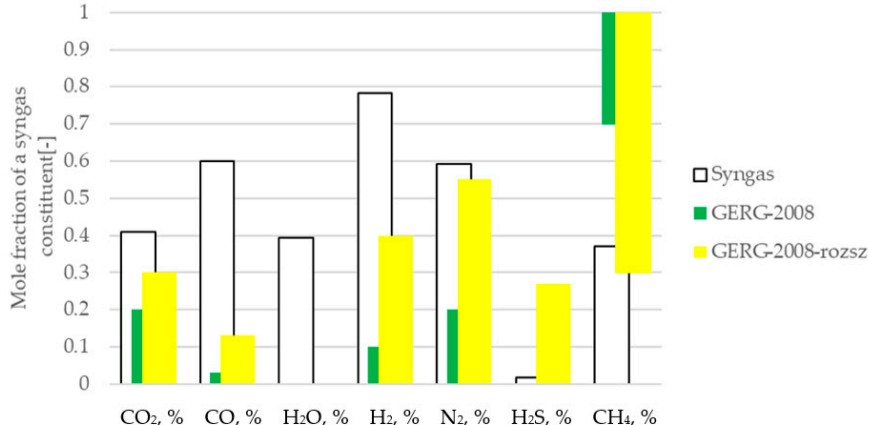

**Figure 2.** The range of mole fractions of the syngas components derived from modeling compared to the scope of the GERG-2008 method in the normal (GERG-2008) and extended (GERG-2008-rozsz) range of composition.

In addition to the methods mentioned above dedicated to NG, other methods for determining the gas state, including syngas (e.g., the Redlich-Kwong equation, the Peng-Robinson equation, and others) [37,40] are known. Those methods rely on values of critical properties (critical temperature and critical pressure) of analyzed gas, which often requires to apply several mixing rules based on the gas composition. There is also an additional difficulty in finding cubic roots of those equations of state in order to obtain the compressibility factor. For this work, the simplified model [36] was used that was confirmed to be effective in gases composed mostly of non-hydrocarbon components. This approach, (a.k.a. the method of weighting treatment), is based on the extraction of the $Z$-factors for both the hydrocarbon and non-hydrocarbon components. This is followed by adding them up, through a weighting treatment, to estimate the final value of the Z-factor for the gas mixture. Since the compressibility factor of all constituents of the obtained syngas is known, this method was chosen to be used for determination of the Z-factor of the obtained syngas. The equation used in this study is presented below (based on [36]):

$$Z_s = \sum_{i=0}^{n} y_i \cdot Z_i \tag{3}$$

where:

$Zs$—compression factor syngas with a given chemical composition,
$yi$—mole fraction of the syngas component,
$Zi$—compressibility factor of syngas component (mono-component gas).

In the case of syngas resulting from the modeling of the SS gasification process, Equation (3) takes the form:

$$Z_s = y_{CO_2} \cdot Z_{CO_2} + y_{CO} \cdot Z_{CO} + y_{H_2O} \cdot Z_{H_2O} + y_{H_2} \cdot Z_{H_2} + y_{CH_4} \cdot Z_{CH_4} + y_{N_2} \cdot Z_{N_2} + y_{H_2S} \cdot Z_{H_2S} \tag{4}$$

The underlying assumptions of the above model are:

- the analyzed gas is treated as a mixture of one-component real gases,
- this model does not take into account phase changes occurring as a result of gas heating or compression,
- it is assumed that in the tested range the mixture is in a gaseous or supercritical state,
- compressibility factor values for mono-component gases are determined for a specific gas state, which corresponds to the modeled state of syngas,
- the gas analyzed under STP conditions behaves like an ideal gas.

Values of compressibility of the mono-constituent gases for specific conditions were determined by the use of the Coolprop software [41], which is used to determine the state of mono-constituent gases. Resulting compressibility values are presented in Table 4.

**Table 4.** The compressibility values of the mono-component gases for the selected pressure.

| $T = 293$ K | $CO_2$ | CO | $H_2O$ | $H_2$ | $CH_4$ | $N_2$ | $H_2S$ |
|---|---|---|---|---|---|---|---|
| $P = 10$ MPa | 0.2107 | 0.9905 | 0.0738 | 1.0608 | 0.8400 | 1.0010 | 0.1732 |
| $P = 15$ MPa | 0.2995 | 1.0074 | 0.1104 | 1.0925 | 0.8000 | 1.0210 | 0.2565 |
| $P = 20$ MPa | 0.3853 | 1.0370 | 0.1467 | 1.1248 | 0.8100 | 1.0517 | 0.3382 |

For modeling of the syngas parameters in a compressed form, the following parameters were assumed. The syngas transport will take place in a mobile storage unit, equipped with tanks with a total capacity of 14,960 m$^3$, with a working pressure of 20 MPa. To determine the gas volume in the transport container, the following Formula (5) was used that was derived from Equation (2) assuming that there is no change in the number of moles of gas under compression:

$$V_2 = \frac{P_1}{P_2} \cdot \frac{Z_2}{Z_1} \cdot \frac{T_2}{T_1} \cdot V_1 \tag{5}$$

where:

$V_2$—the volume of gas in a transport unit, nm$^3$,
$P_1$—working pressure of the transport unit, MPa,
$Z_1$—compressibility coefficient of gas at temperature $T_1$, pressure $P_1$ and volume $V_1$,
$V_1$—water volume of the transport unit, m$^3$,
$P_2$—gas pressure under STP conditions, MPa,
$Z_2$—gas compressibility coefficient under STP conditions,
$T_1$, $T_2$—gas temperatures in state 1 and 2.

Assuming that the real gas could be considered as ideal under STP conditions (273.15 K and absolute pressure of 100 kPa) and taking into account boundary conditions, Equation (5) may have a form (where t denotes the temperature of the gas in transport, in K):

$$V_2 = \frac{P_1}{0.1} \cdot \frac{1}{Z_1} \cdot \frac{273.15}{t} \cdot V_1 \tag{6}$$

Assumption of the ideality of the gas under STP conditions is in agreement with work performed by Werle [32] who used similar assumptions for the modeling of sewage sludge gasification. This made it possible to refer results obtained in this manuscript to other researches. After the simplification of (6), $V_2$ may be calculated as follows:

$$V_2 = 10 \cdot \frac{P_1}{Z_1} \cdot \frac{273.15}{t} \cdot V_1 \tag{7}$$

In the case of natural gas, the ratio of the water volume of the transport unit to the gas volume under STP conditions is designated as $Bg$ (FVF) (formation volume factor) [38]. In this paper, the authors defined a similar coefficient for the syngas, which was defined as the inverse of FVF and referred to as the syngas compression ratio:

$$B_{syn} = \frac{V_2}{V_1} = 10 \cdot \frac{P_1}{Z_1} \cdot \frac{273}{T} \tag{8}$$

The resulting volume of the syngas in the transport can be determined using a simplified form of Equation (8):

$$V_2 = B_{syn} \cdot V_1 \tag{9}$$

The value of the $B_{syn}$ coefficient describes, how many times the volume of the syngas will increase when compressed to a given pressure, here equal to 20 MPa.

For the assumed boundary conditions of the analysis ($P_1$ = 20 MPa, $T$ = 293 K), Equation (10) may be written:

$$B_{syn} = 186.3481 \cdot Z_1^{-1} \tag{10}$$

From Equation (10), it follows that as the gas compressibility factor increases, the value of the syngas compression coefficient decreases. The amount of energy in the transport unit was determined by the formula:

$$E = V_2 \cdot LHV = B_{syn} \cdot V_1 \cdot LHV \tag{11}$$

For the assumed volume of the water in the transport unit and taking Equation (11) into account, Equation (12) has a form:

$$E = B_{syn} \cdot V_1 \cdot LHV \cong 2.75 \cdot \frac{LHV}{Z_1} \tag{12}$$

From formula (12), it follows that with the increase of the calorific value of the gas obtained and as the gas compressibility coefficient decreases, the energy content in a given volume of a mobile storage unit under defined pressure increases.

To sum up, the modeling technique in this study utilizes a real gas equation to calculate the volume of gas, that will occupy space in STP conditions after its decompression from a pressure vessel in which this gas occupies 14,960 m$^3$ of volume under pressure of 20 MPa. The calculation scheme involved implementation of the given syngas composition in Equation (4) to determine the $Z$ factor and then use this $Z$ factor to determine the volume of gas in the transport, $B_{syn}$, and energy content from Equations (7), (9) and (11), respectively. These calculations were repeated five times for five different syngas compositions.

## 3. Results

Results presented in this chapter are structured in two subchapters. In the first one, the influence of the molar composition of the gasifying agents and process temperature on *LHV* is presented. Figures 3–7 present a linear approximation of the obtained results and corresponding equations that makes it possible to calculate *LHV* for a given gasifying agent, molar concentration (x) and process parameter (y). Obtained results were statistically analyzed using multiple regression to identify the influence of process parameters on the syngas *LHV*. In the second subsection, the influence of the molar composition of the gasifying agents and process temperature on the syngas compression ratio and energy content in the transport is presented. Results are presented for each gasifying agent as follows:

- $CO_2$, Figures 8–10;
- $H_2O$, Figures 11–13;
- Air, Figures 14–16;
- $O_2$, Figures 17–19;
- H2, Figures 20–22.

Figures 8, 11, 14, 17 and 20 present a linear approximation of the obtained results of the $B_{syn}$ coefficient for each gasifying agent. Figures 9, 12, 15, 18 and 21 present a linear approximation of the obtained results of energy content in the transport of syngas in pressure vessels with a total capacity of 14,960 m$^3$ at 200 bar at 293 K. Figures 10, 13, 16, 19 and 22 present the theoretical value of the compressibility coefficient $Z$ obtained using each gasifying agent.

*3.1. Modeling the SS Gasification Process—Effects of $O_2$, $H_2$, $CO_2$, Water Vapor and Air; Temperature and Concentration of the Gasifying Agent on LHV*

The heating values of the produced syngas with the usage of different gasification agents differed from each other. When the SS gasification was with pure oxygen the resulting calorific value of the

syngas ranged from 17.20 MJ·(Nm$^3$)$^{-1}$ (for 0.1 mole oxygen per mole of carbon at the temperature of 973 K) to 2.80 MJ·(Nm$^3$)$^{-1}$ (for 1 mole oxygen per mole of carbon at a temperature of 973 K), respectively. The average calorific value of the produced syngas obtained regardless of the amount of the oxygen fed was 8.60 MJ·(Nm$^3$)$^{-1}$ at 1123 K (Figure 3).

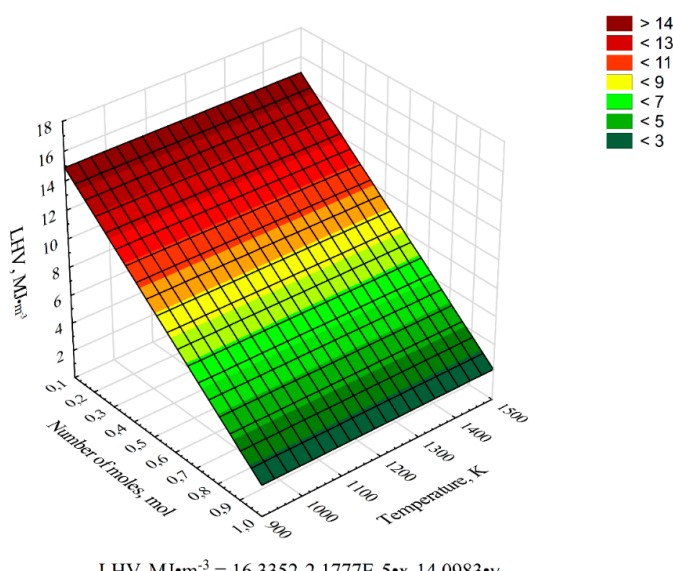

LHV, MJ•m$^{-3}$ = 16.3352-2.1777E-5•x-14.0983•y

**Figure 3.** The effect of the number of moles of the gasification agent oxygen (x) and temperature (y) on the heating value (*LHV*) of syngas.

During gasification with hydrogen, the calorific values of the syngas were in the range of 20.40 MJ·(Nm$^3$)$^{-1}$ at 0.4 mole hydrogen per mole of carbon and process temperature 973 K to 13.4 MJ·(Nm$^3$)$^{-1}$ for four mole hydrogen per mole of carbon at a temperature of 1198–1273 K (Figure 4). The average calorific value of the gas was 15.70 MJ·(Nm$^3$)$^{-1}$ at an average process temperature of 1123 K.

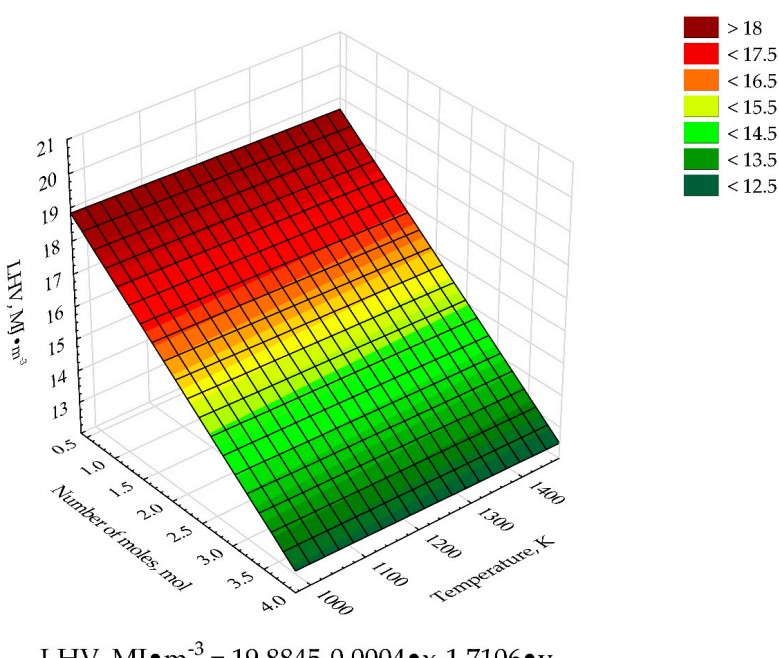

LHV, MJ•m$^{-3}$ = 19.8845-0.0004•x-1.7106•y

**Figure 4.** Effect of the number of moles of the gasification agent hydrogen (x) and temperature (y) on the heating value (*LHV*) of syngas.

When the gasification agent was $CO_2$, the gas calorific values ranged from 16.60 MJ·$(Nm^3)^{-1}$ (at a process temperature of 973 K for 0.2 moles of $CO_2$ per mole of carbon) to 7.5 MJ·$(Nm^3)^{-1}$ (for two moles of $CO_2$ per mole of carbon and process temperature 973–998 K) (Figure 5), respectively. The average calorific value of the gas was 10.40 MJ·$(Nm^3)^{-1}$ for the average process temperature at 1123 K.

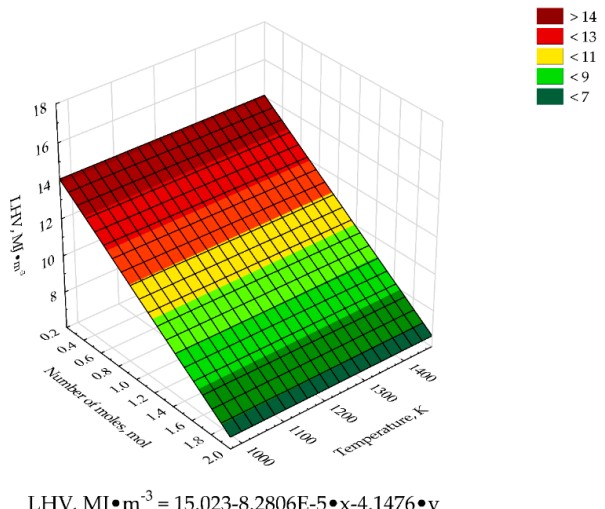

LHV, MJ•m$^{-3}$ = 15.023−8.2806E-5•x−4.1476•y

**Figure 5.** The effect of the number of moles of the gasification agent carbon dioxide (x) and temperature (y) on the heating value (*LHV*) of syngas.

The results for gasification with steam were very similar to those obtained for carbon dioxide. *LHV* ranged from a maximum of 16.50 MJ·$(Nm^3)^{-1}$ (for 0.2 moles of water molecules per mole of carbon at 973 K) to a minimum of 6.9 MJ·$(Nm^3)^{-1}$ (for two moles of water molecules per mole of carbon and process temperature in the range of 973 K–998 K), respectively. The average calorific value of gas was 9.9 MJ·$(Nm^3)^{-1}$ for the average temperature of 1123 K (Figure 6).

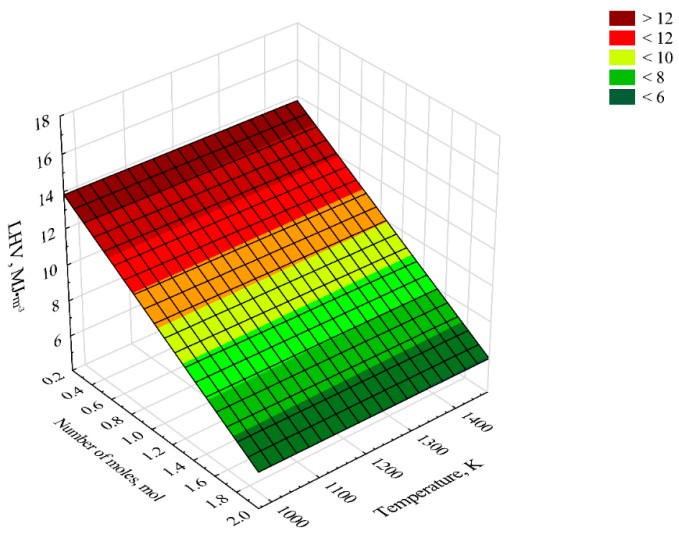

LHV, MJ•m$^{-3}$ = 14.8871−0.0002•x−4.3409•y

**Figure 6.** Effect of the number of moles of the gasification agent water vapor (x) and temperature (y) on the heating value (*LHV*) of synthesis gas.

The last examined gasification agent was air, for which the lowest *LHV* of the produced gas was achieved, i.e., it ranged from 13.40 MJ·$(Nm^3)^{-1}$ (at 973 K for 0.1 moles of oxygen per mole of carbon and

0.367 moles of nitrogen per mole of carbon) to the lowest value of 0.90 MJ·$(Nm^3)^{-1}$ (per one mole of oxygen per mole of carbon and 3.76 moles of nitrogen per mole of carbon, and the process temperature was in the range of 973–1048 K) (Figure 7), respectively.

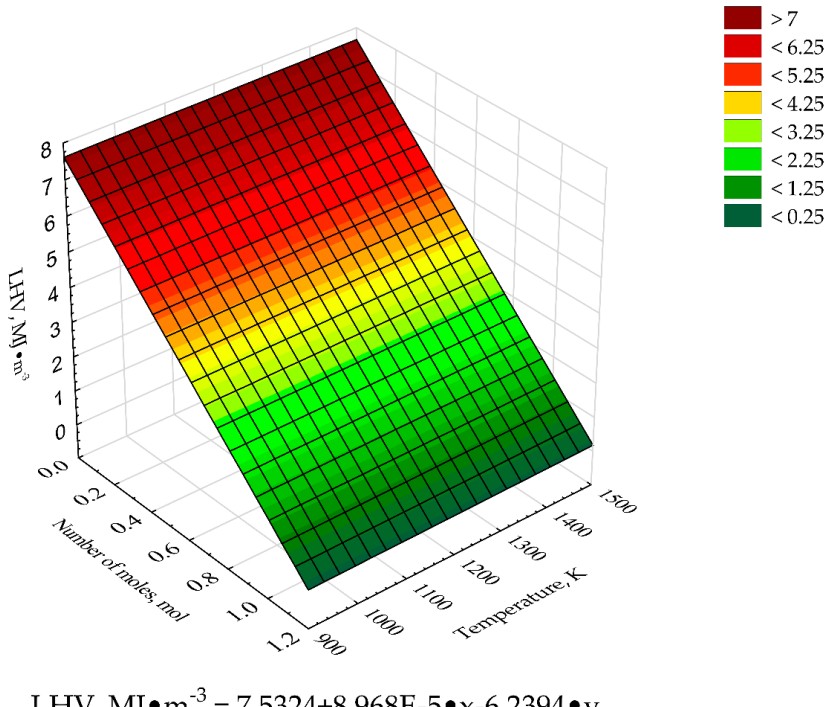

$$LHV, MJ \bullet m^{-3} = 7.5324 + 8.968E\text{-}5 \bullet x - 6.2394 \bullet y$$

**Figure 7.** The effect of the number of moles of the gasification agent air (x) and temperature (y) on the heating value (*LHV*) of syngas.

During gasification, the amount of the supplied gasification agent is a very important parameter, because it significantly affects the obtained calorific value of the produced syngas. This was demonstrated by the multiple regression analysis carried out with respect to independent variables (the number of molar ratios of gasification agents to carbon in the SS and temperature). Only in the case of the gasification analysis carried out using air as a gasifying agent, the temperature had a significant impact on the obtained calorific value of synthesis gas (Table 5). This was confirmed by the probability value, which was lower than 0.05.

**Table 5.** Summary of parameterization evaluation of variables independent of the calorific value of syngas.

| Gasification Agents | Factor | Statistical Parameter | | | | | |
|---|---|---|---|---|---|---|---|
| | | Standardized Regression Coefficient | The Standard Error of the Standardized Regression Coefficient | Regression Coefficient | The Standard Error of the Regression Coefficient | Value of Statistic at $\alpha < 0.95$ and the Degree of Freedom t (t = 57) | A Probability Value ($p$) |
| | | | | **LHV** | | | |
| Carbon dioxide | Intercept | - | - | 15.023 | 0.574 | 26.141 | 0.000 |
| | Molar ratios of gasification agents to carbon | −0.923 | 0.026 | 4.147 | 0.119 | −34.671 | 0.000 |
| | Temperature, K | −0.004 | 0.026 | 0.000 | 0.000 | −0.182 | 0.855 |
| Water vapor | Intercept | - | - | 1.887 | 0.599 | 24.831 | 0.000 |
| | Molar ratios of gasification agents to carbon | −0.924 | 0.026 | −4.340 | 0.124 | −34.784 | 0.000 |
| | Temperature, K | −0.010 | 0.026 | 0.000 | 0.000 | −0.396 | 0.692 |
| Air | Intercept | - | - | 7.532 | 0.046 | 161.851 | 0.000 |
| | Molar ratios of gasification agents to carbon | −0.998 | 0.003 | −6.239 | 0.019 | −322.030 | 0.000 |
| | Temperature, K | 0.007 | 0.003 | 0.001 | 0.000 | 2.439 | 0.015 |
| Oxygen | Intercept | - | - | 16.335 | 0.494 | 33.010 | 0.000 |
| | Molar ratios of gasification agents to carbon | −0.978 | 0.014 | −14.098 | 0.206 | −68.432 | 0.000 |
| | Temperature, K | −0.001 | 0.014 | 0.000 | 0.000 | −0.055 | 0.955 |
| Hydrogen | Intercept | - | - | 19.884 | 0.428 | 46.436 | 0.000 |
| | Molar ratios of gasification agents to carbon | −0.936 | 0.024 | −1.710 | 0.044 | −38.382 | 0.000 |
| | Temperature, K | −0.027 | 0.024 | 0.000 | 0.000 | −1.108 | 0.268 |

### 3.2. Modeling of the Syngas Compression Process

The results of modeling of the syngas compression process with the simulated compositions are presented in the form of charts of the $B_{syn}$ coefficient value and the energy content in the transport unit in relation to the gasification temperature and the molar ratio of gasification agents and carbon in SS. The highest value of the $B_{syn}$ of 404.61 was observed for the gasification of SS with oxygen at a temperature of 1473 K while the smallest value equaled 175.23 was for the gasification with hydrogen at the same temperature. This allows to state that the gasification agent type has an effect on the value of the $B_{syn}$ coefficient, and hence on the syngas content in the gas transport unit.

In the case of the gasification of SS with the use of carbon dioxide, the highest value of the $B_{syn}$ coefficient was 254.10 for two moles of $CO_2$ per mole of carbon at 1473 K. The smallest coefficient value was obtained for 0.4 moles of $CO_2$ per mole of carbon at 1473 K. The highest energy content in the transport unit was 47,436.54 MJ for 0.2 moles of $CO_2$ per mole of carbon at 973 K, the lowest for two moles of $CO_2$ per mole of carbon at 998 K (Figures 8–10).

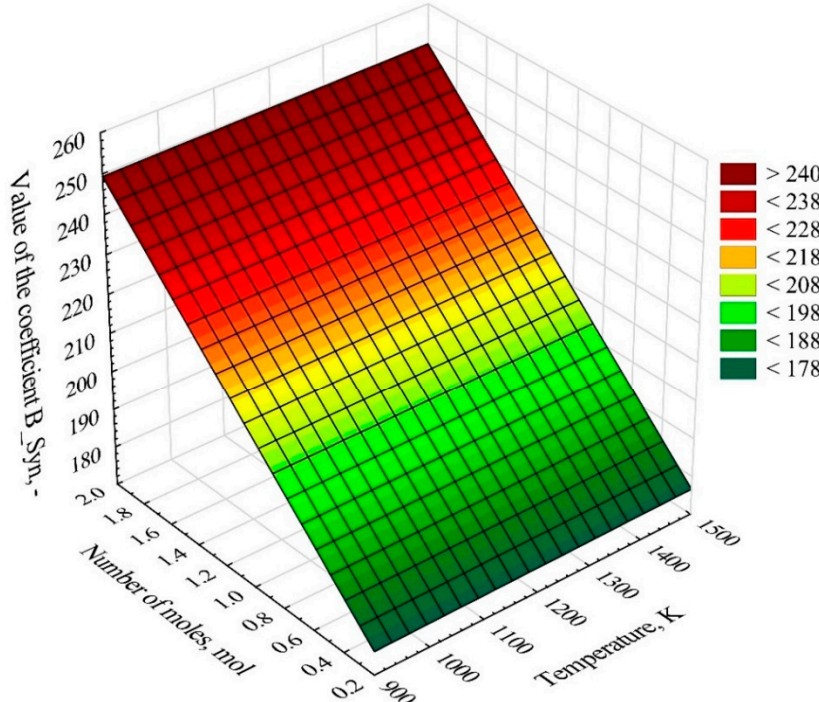

Value of the coefficient B_Syn, - = 167.1549+0.0009·x+40.6928·y

**Figure 8.** Value of the syngas compression ratio as a function of the number of moles of the gasification agent (x) and the temperature (y) of the gasification process using $CO_2$.

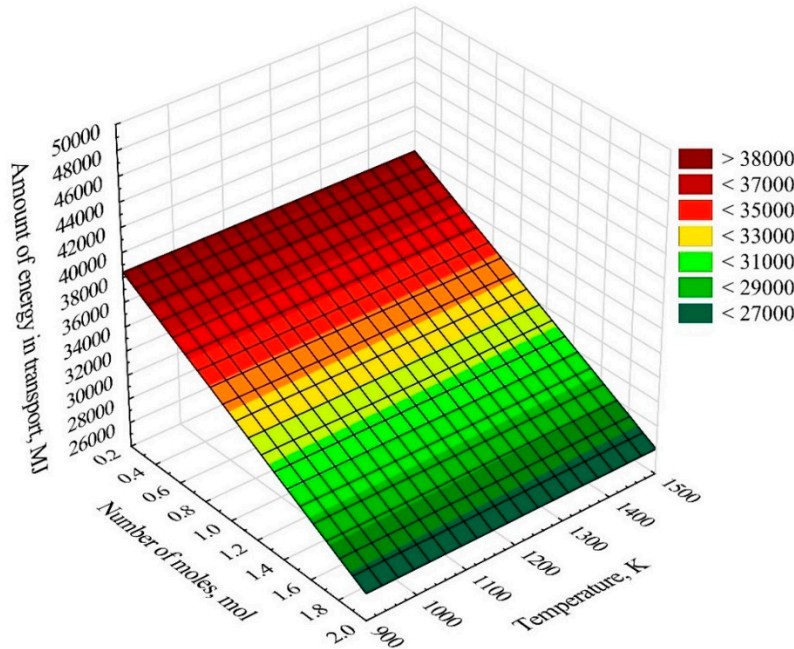

Amount of energy in transport, MJ = 39992.5649-0.2215•x-6787.334•y

**Figure 9.** Energy content in the transport of syngas in the pressure vessels with a capacity of 14,960 m$^3$ at 200 bar at 293 K as a function of the number of moles of the gasification agent (x) and the temperature (y) of the gasification process using $CO_2$.

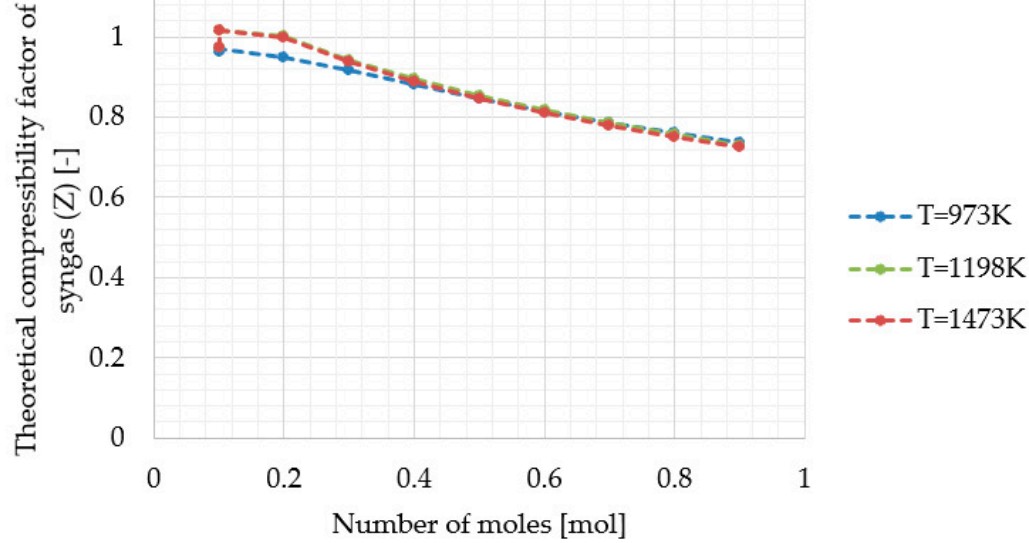

**Figure 10.** The theoretical value of compressibility coefficient Z from the gasification of $CO_2$.

In the case of gasification of SS with $H_2O$, the largest value of the $B_{syn}$ coefficient was 257.35 for two moles of $H_2O$ per mole of carbon at 1473 K. The smallest value of 178.13 was obtained for 0.4 moles of $H_2O$ per mole of carbon at 1473 K. The highest energy content in the transport unit was 47,013.92 MJ for 0.2 moles of $H_2O$ per mole of carbon at 973 K, the smallest being 26,291.12 MJ for two moles of $H_2$ per mole of carbon at 1023 K (Figures 11–13).

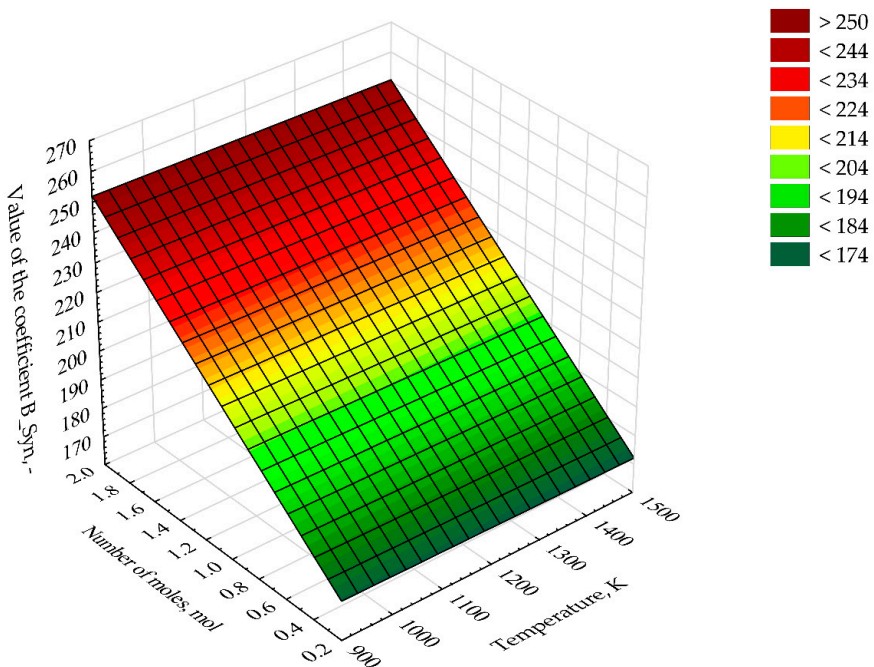

Value of the coefficient B_Syn, - = 166.7992-0.0022•x+43.586•y

**Figure 11.** Value of the syngas compression ratio as a function of the number of moles of the gasification agent (x) and the temperature (y) of the gasification process using $H_2O$.

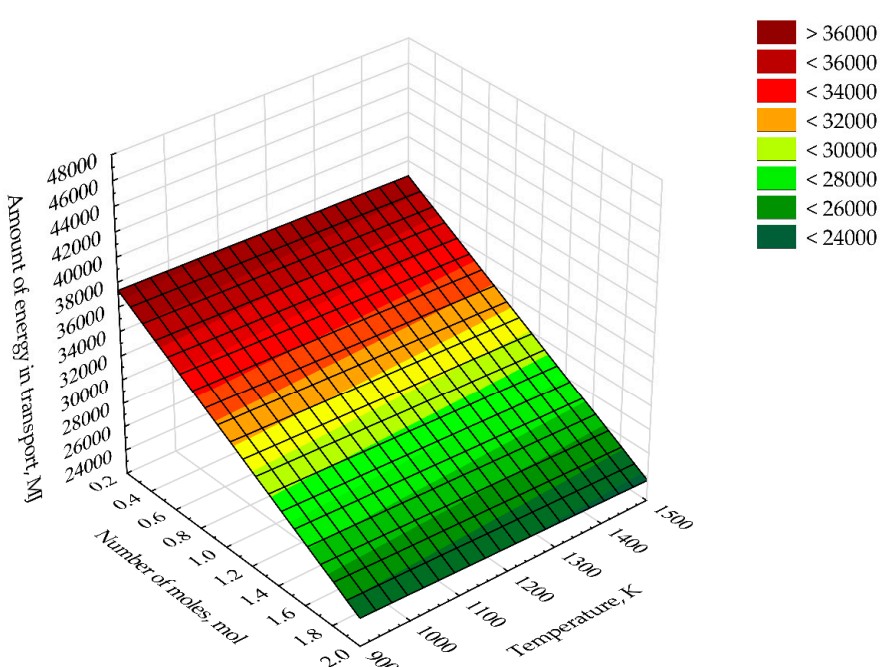

Amount of energy in transport, MJ = 39693.1829-1.0103•x-7230.5389•y

**Figure 12.** Energy content in the transport of syngas in the pressure vessels with a capacity of 14.960 $m^3$ at 200 bar at 293 K as a function of the number of moles of the gasification agent (x) and the temperature (y) of the gasification process using $H_2O$.

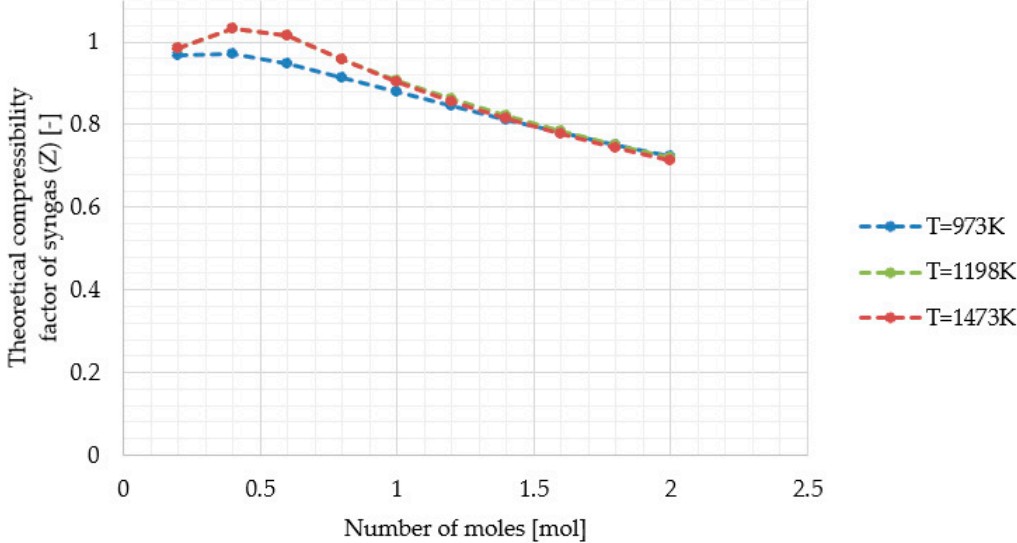

**Figure 13.** The theoretical value of the compressibility coefficient Z from the gasification of $H_2O$.

For the gasification of SS with air, the highest value of the $B_{syn}$ coefficient was 262.26 for one mole of oxygen from air per mole of carbon at 1473 K. The smallest value of the coefficient of 184.26 was obtained for 0.2 mole of oxygen from air at 1473 K. The highest energy content in the transport unit was 19,916.74 MJ for 0.1 moles of $H_2O$ per mole of carbon at 973 K, the smallest being 5,508.56 MJ for one mole of oxygen from air per mole of carbon at 973 K (Figures 14–16).

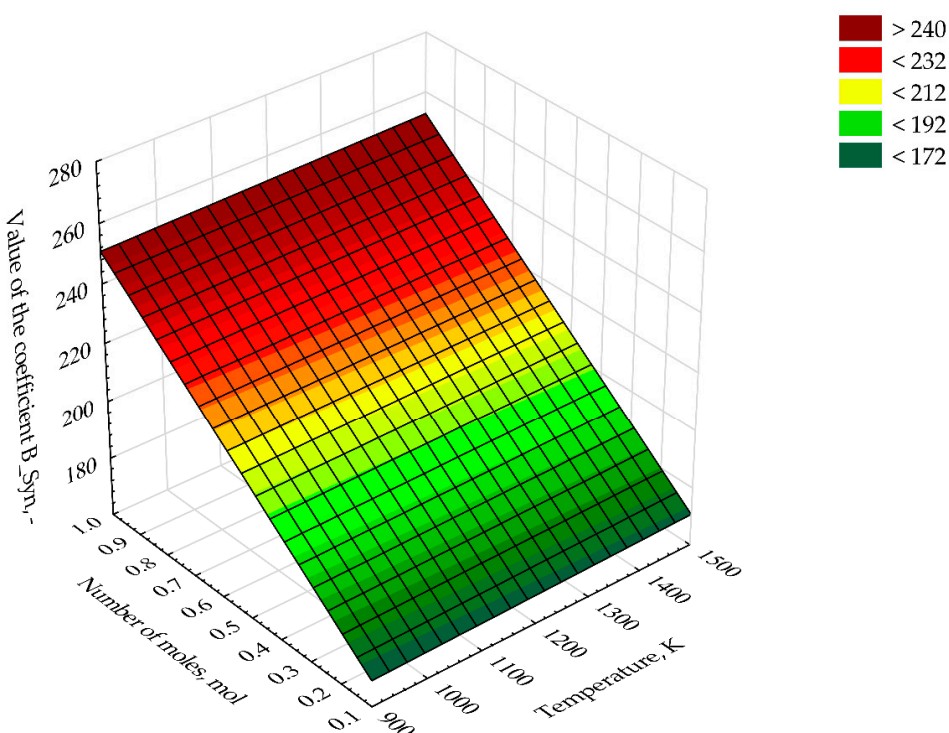

Value of the coefficient B_Syn, - = 157.2017+0.003•x+90.8996•y

**Figure 14.** Value of the syngas compression ratio as a function of the number of moles of the gasification agent (x) and the temperature (y) of the gasification process using air.

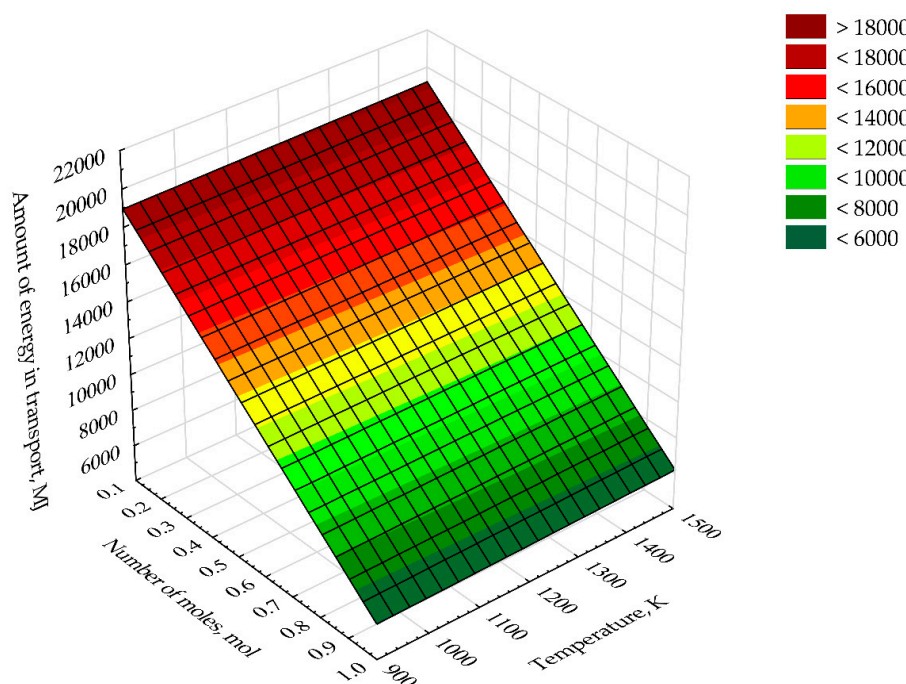

Amount of energy in transport, MJ = 20031.5766+0.4099•x−14445.1502•y

**Figure 15.** Energy content in the transport of syngas in the pressure vessels with a capacity of 14.960 m$^3$ at 200 bar at 293 K as a function of the number of moles of the gasification agent (x) and the temperature (y) of the gasification process using air.

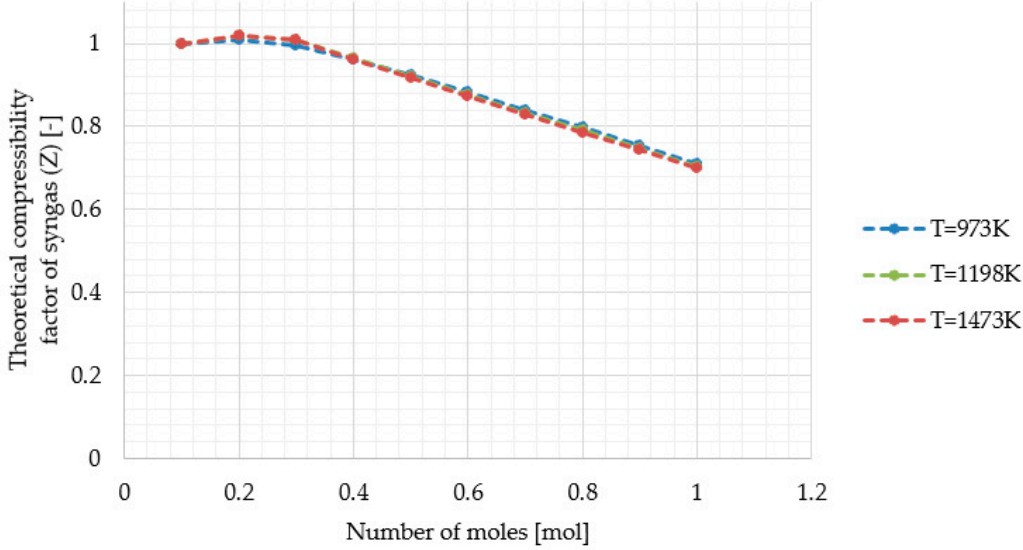

**Figure 16.** The theoretical value of the compressibility coefficient Z from the gasification of air.

During the gasification of SS with oxygen the largest value of the $B_{syn}$ coefficient was 404.61 for one mole of oxygen per mole of carbon at 1473 K. The smallest value of 175.27 was obtained for 0.2 moles of oxygen per mole of carbon at 1473 K. The highest energy content in the transport unit was 47,909.84 MJ for 0.1 moles of oxygen per mole of carbon at a temperature of 973 K, the lowest being 16,025.41 MJ for one mole of oxygen per mole of carbon at 973 K (Figures 17–19).

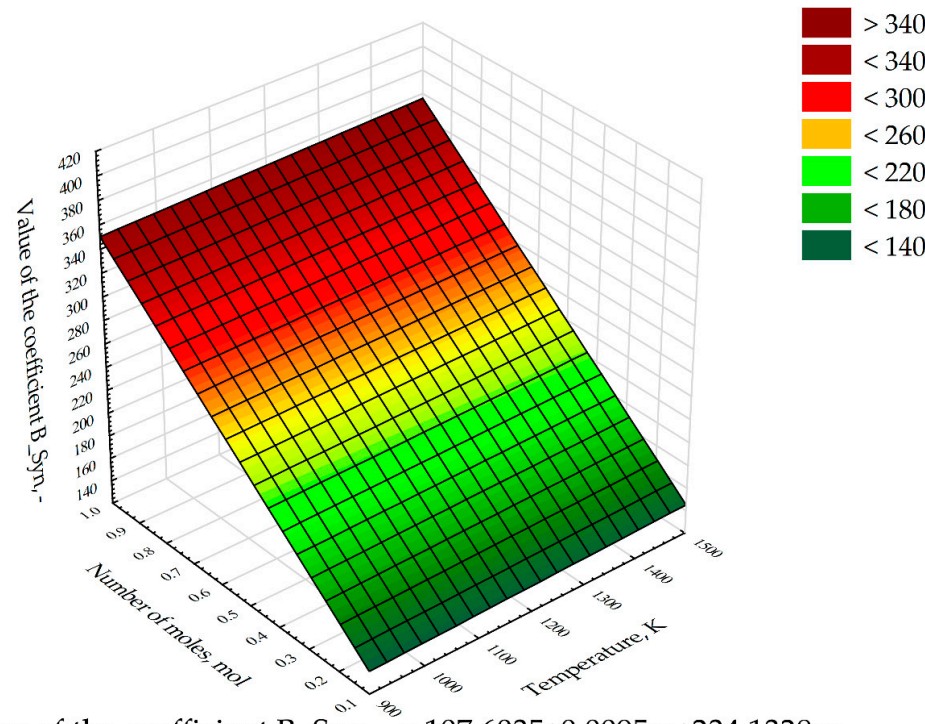

Value of the coefficient B_Syn, - = 107.6035+0.0095•x+234.1328•y

**Figure 17.** Value of the syngas compression ratio as a function of the number of moles of the gasification agent (x) and the temperature (y) of the gasification process using $O_2$.

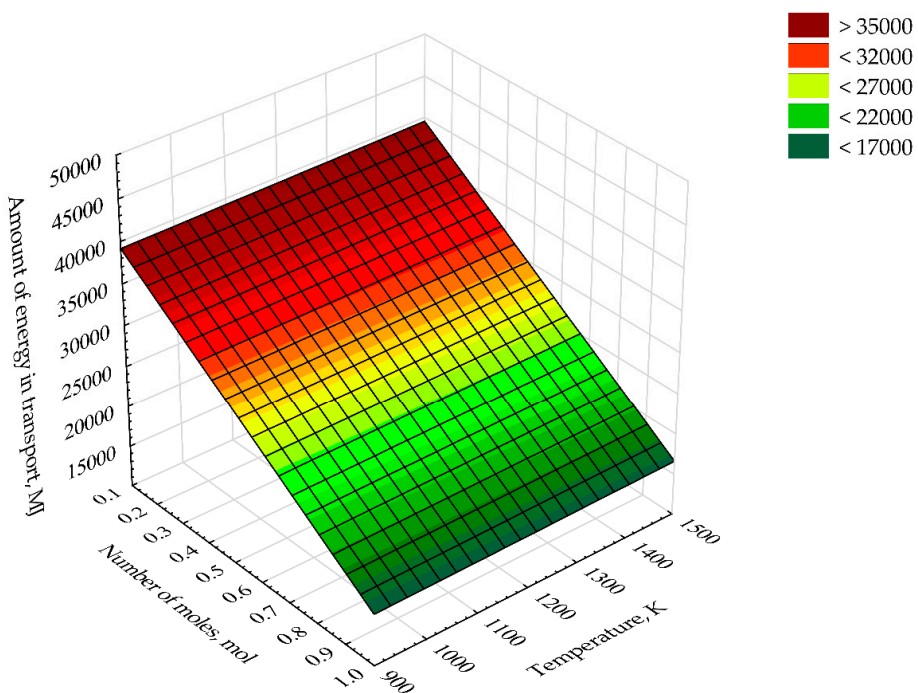

Amount of energy in transport, MJ = 41170.6472+0.6011•x-25457.7947•y

**Figure 18.** Energy content in the transport of syngas in the pressure vessels with a capacity of 14.960 $m^3$ at 200 bar at 293 K as a function of the number of moles of the gasification agent (x) and the temperature (y) of the gasification process using $O_2$.

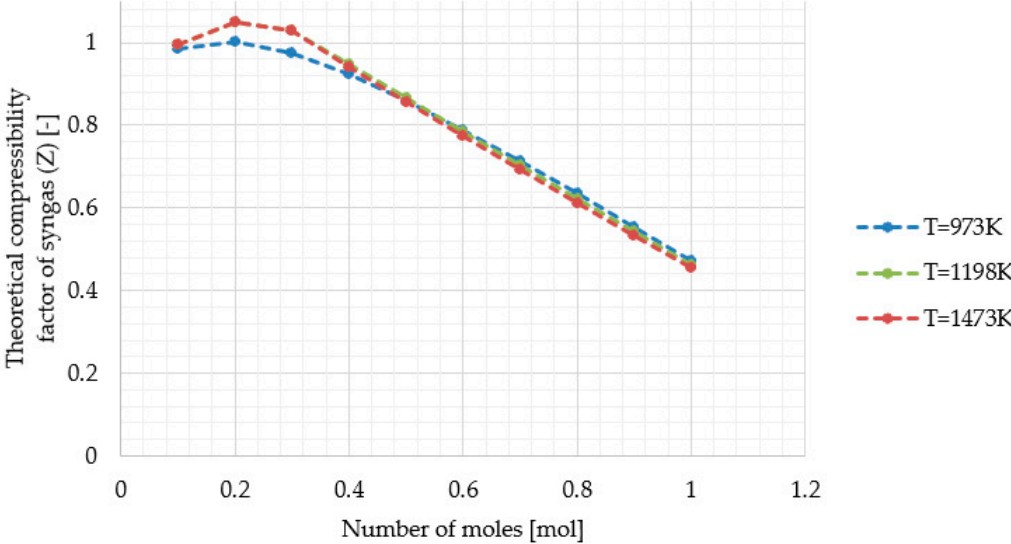

**Figure 19.** The theoretical value of the compressibility coefficient Z from the gasification of $O_2$.

For the gasification of SS with hydrogen, the largest value of the $B_{syn}$ coefficient was 196.84 for 0.4 moles of hydrogen per mole of carbon at 973 K. The smallest value of the 175.23 coefficient was obtained for four moles of hydrogen per mole of carbon at 1473 K. The highest energy content in the transport unit was 65,296.98 MJ for 0.4 moles of hydrogen per mole of carbon at 973 K, the smallest being 38,227.50 MJ for four moles of hydrogen per mole of carbon in 1473 K (Figures 20–22).

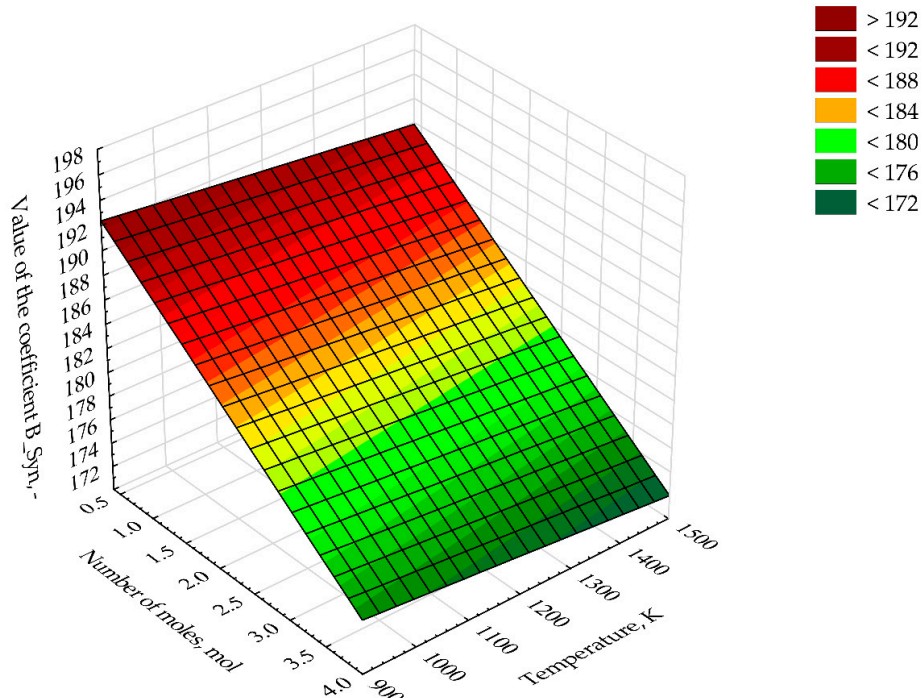

Value of the coefficient B_Syn, - = 198.12-0.0042•x-4.9614•y

**Figure 20.** The value of the syngas compression ratio as a function of the number of moles of the gasification agent (x) and the temperature (y) of the gasification process using $H_2$.

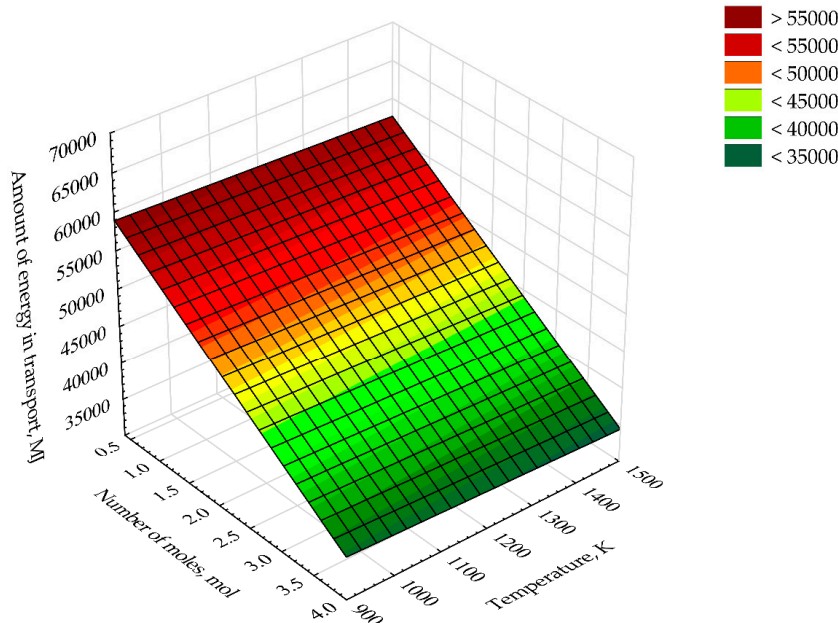

Amount of energy in transport, MJ = 63390.8696-2.1265•x-6447.4056•y

**Figure 21.** Energy content in the transport of syngas in the pressure vessels with a capacity of 14,960 m$^3$ at 200 bar at 293 K as a function of the number of moles of the gasification agent (x) and the temperature (y) of the gasification process using H$_2$.

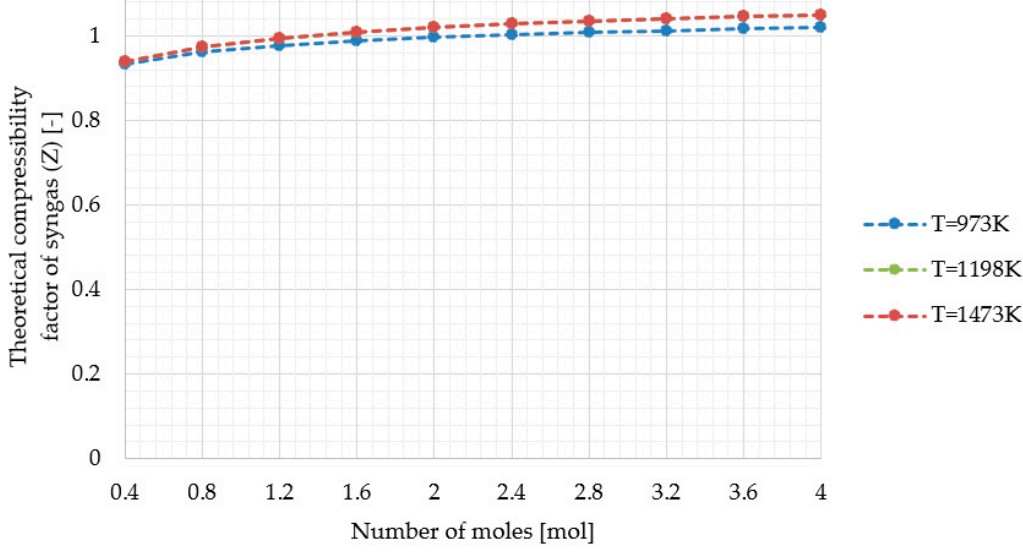

**Figure 22.** The theoretical value of the compressibility coefficient Z syngas derived from gasification with hydrogen.

## 4. Discussion

The amount of the gasification agent is reflected in the calorific value of syngas. The obtained results are consistent with the theory [5] and the C-H-O diagram in the gasification process. Additionally, the gas heating values that depend on the gasification agent correspond to the results presented elsewhere [5]. The calorific value of syngas from SS, when the gasification agent was the atmospheric air, correspond to tests carried out by [42,43] where syngas with average calorific values of 2.55, 3.20 and 4.00 MJ·(Nm$^3$)$^{-1}$ was obtained, respectively. The air used for gasification resulted in the production

of gas with a high N content (24.5% to 66.2% on average), which effectively reduced the calorific value of syngas, similarly as [44] where the maximum calorific value of the syngas was 6 MJ·(Nm$^3$)$^{-1}$.

In this work, such low values were achieved when the amount of the gasifying agent was greater than 0.3 moles of oxygen per mole of carbon and 1.128 moles of nitrogen per mole of carbon. In studies carried out by [45] during the application of sub-stoichiometric air doses, a gas with a calorific value of 10–11 MJ·kg$^{-1}$ was obtained, which also corresponds to the results obtained in this modeling analysis. In [46], the results obtained in the Gaseq program during the gasification of SS by air were similar to the results obtained in this work. The calorific value of the produced syngas in all cases had a statistically significant decrease with the increase in the amount of the gasification agent supplied.

In all cases, the composition of the syngas was related to the values of the equilibrium of the gasification reaction, which varied depending on the temperature and concentration of the products and substrates [47]. The highest values of the synthesis rate constant have the sequence of reactions [27]:

- carbon with molecular oxygen to form CO,
- steam-carbon with the production of $CO_2$ and $H_2$,
- carbon with $CO_2$ to form CO.

The quoted properties coincide with the syngas composition data obtained from modeling. At a low gasification agent concentration, significant amounts of CO were observed, which was associated with the partial combustion of the substrate with oxygen deficiency. The CO decreased with the increase in oxygen supply, which was caused by the oxidation of CO to $CO_2$, the concentration of which increased in the produced syngas together with the increase in the amount of supplied gasifying agent. In subsequent reactions, the material was further degraded to molecular hydrogen and an additional amount of CO. These reactions were followed by reactions associated with the release of residual water from the gasified material. The hydrogen formed in the first gasification phase was then used together with $CO_2$ for the synthesis of $CH_4$.

As a result of the modeling of the syngas compression process, it was found that the number of moles of all tested gasification agents: $H_2$, $O_2$, air, steam, and $CO_2$ had a significant effect on the energy content in the transport unit, with temperature having only a minor influence. The proposed model based on the method of the weighting treatment was applied to the syngas, and obtained results were similar to [17] in terms of the compression coefficient evaluated using the Peng-Robinson and SRK equations of state for syngas having similar composition to the one used by APT et al. (composition by weight: 0% $CH_4$, 45% CO, 35.4% $H_2$, 17.1% $CO_2$, 2.1% $N_2$, 0.4% $H_2O$ [17]). The application of the method of the weighting treatment was useful in the determination of the syngas compressibility factor, for which the precise composition of the gas is required. The proposed model based on a real gas equation showed, that for each gasifying agent, the highest energy content in the SETS was obtained at a temperature of 973 K, with the highest value obtained for gasification with $H_2$. In the same time, gasification with $H_2$ resulted in the lowest values of the $B_{syn}$ which indicate, that the significant influence on the energy content in the transport unit results from the calorific value of syngas and not the increase of compressibility of syngas at higher pressures (Figure 23).

Another result was, that the bigger the value of $B_{syn}$ the bigger the volume of gas in the transport. What is also a significant result is that the largest energy content, which in fact determines the effectiveness of the proposed syngas energy transportation system (SETS) was obtained for a syngas that did not have the largest $B_{syn}$ coefficient. This is an important insight, i.e., it means, that the optimum configuration for syngas (from the energy content criteria) is not the one in which the most volume of gas can be stored. This is a very important finding indicating the direction of development and optimization of the proposed system.

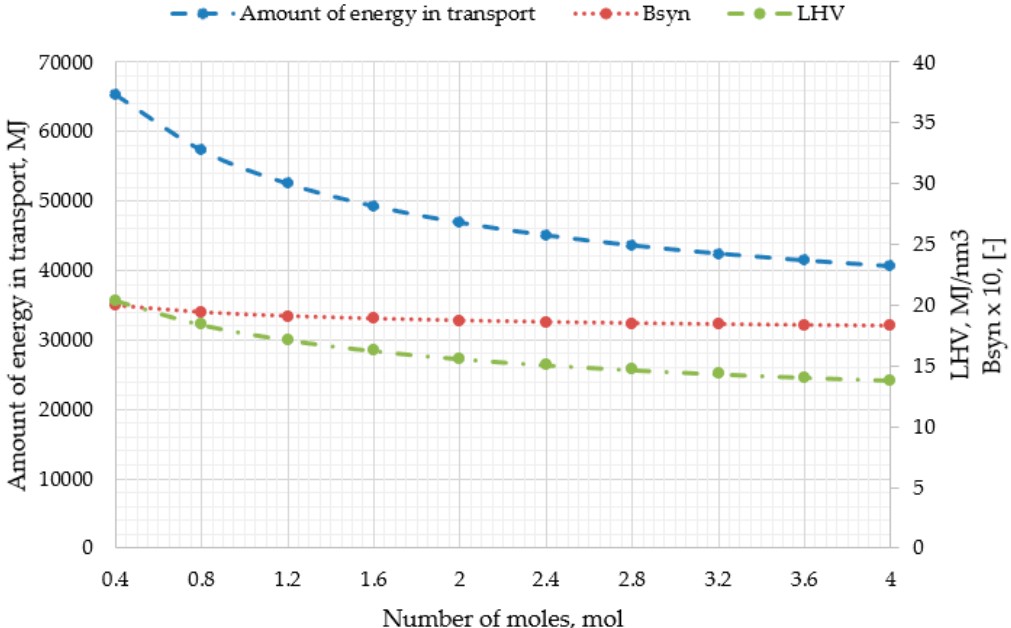

**Figure 23.** Amount of energy in the transport, *LHV*, and $B_{syn}$ coefficient for syngas obtained as a result of gasification using $H_2$ at a temperature of 973 K.

For all the gasifying agents except $H_2$, the increase in the $B_{syn}$ coefficient is observed with the increase in the number of moles of the gasification agent (Figure 24). This is due to the increasing percentage in the syngas composition of gases ($O_2$, $CO_2$, $H_2O$), which have lower values of compressibility factors Z. Since these gases do not have high calorific values, the total calorific value of the syngas in the transport unit decreases with the increase in the number of gas moles even though it is possible to transport more gas in the same transport unit. The largest value of the $B_{syn}$ coefficient has been obtained for gasification with $O_2$, which means that when gasifying with this gas, it is possible to transport the largest gas volume of the syngas in a given volume.

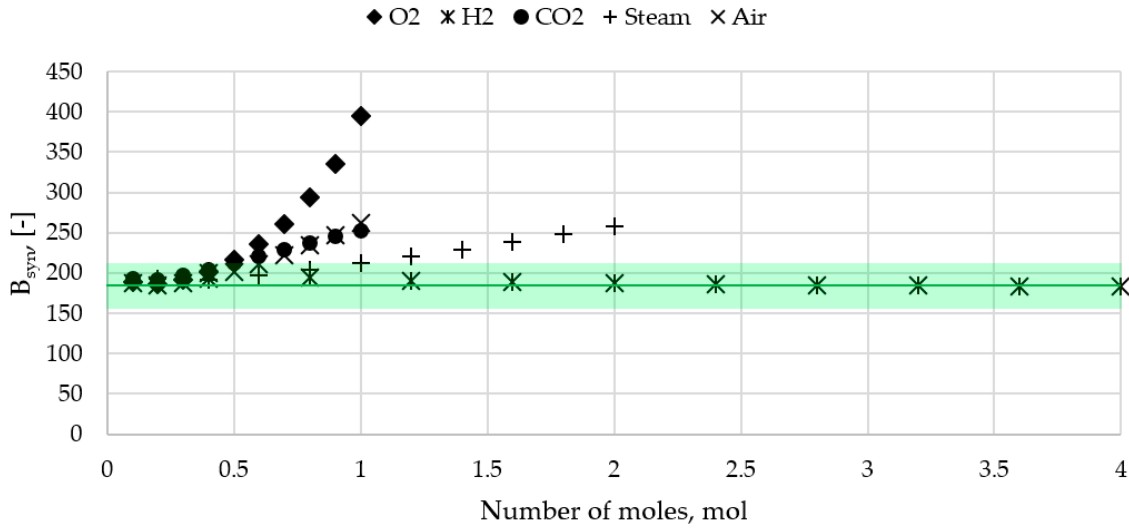

**Figure 24.** The $B_{syn}$ value for various gasifying agents and the area of applicability of an ideal gas equation with the area of 15% deviation from the value of $B_{syn}$ = 186.3481.

It was also observed that the value of the syngas compressibility factor, which for specific ranges of the gasification agent concentration (less than one mole for $CO_2$ and $H_2O$, less than 0.5 for air

and in the range of 0.4 to 4 for $H_2$) is very close to one (maximum deviation in the given ranges is approximately 15%). This means that in the given ranges, the syngas compressed to 20 MPa at 293 K can be treated as an ideal gas, which confirms the results of a study published in [17]. Above the given ranges, there is a significant decrease in the syngas compression ratio, which may indicate a potential risk of a multiphase state which is not included in the developed model. Therefore, for concentrations of the gasification agents above the scope of applicability of the ideal gas equation, it is necessary to model the compression process based on models that take into account the multiphase structure of the compressed gases. As a result of this study, the maximum energy content in the transport was 65,296.98 MJ for the syngas obtained using $H_2$ as a gasifying agent. This corresponds to the transport of 2944.72 $nm^3$ of the syngas at 20 MPa in 14,960 $m^3$. Table 6 summarizes the resulting energy content, the value of $B_{syn}$, and volume of gas in the transport gasifying temperature and molar concentration of a given gasifying agent.

**Table 6.** Presentation of results of the study. Energy content in the transport for given process parameters.

| Gasifying Agent | Parameters of the Gasification Process | | Parameters of Pressurized Storage | |
|---|---|---|---|---|
| | No. of Moles | Temperature, K | Volume in Transport,$nm^3$ | Energy Content in Transport, MJ |
| $CO_2$ | 0.2 | 973 | 2832.72 | 47,436.54 |
| $H_2O$ | 0.2 | 973 | 2847.59 | 47,013.92 |
| Air | 0.1 | 973 | 2758.73 | 19,916.74 |
| $O_2$ | 0.1 | 973 | 2791.88 | 47,909.84 |
| $H_2$ | 0.4 | 973 | 2944.72 | 65,296.98 |

Assuming that an average house (residential dwelling) uses 500 $m^3$ $year^{-1}$ of NG in the heating season in Poland having an energy content of 35 MJ $(nm^3)^{-1}$, the estimated energy demand is ~17,500 MJ $year^{-1}$. Thus, the syngas obtained as a result of the gasification process conducted at 973 K for each gasifying agent, compressed to 20 MPa and stored in the pressure vessel of 14,960 $m^3$ can provide enough energy to meet this demand. Obtained results prove, therefore, that the compressed syngas can be a valuable fuel that can increase the diversity of fuels used, increasing the share of renewable energy sources worldwide. At the same time, the proposed SETS makes it possible to deliver the compressed syngas to remote areas, which further increases the availability of renewable sources.

## 5. Conclusions

Modeling tests have shown that the selection of the gasifying agent affects the composition and calorific value of the syngas from the sewage sludge. It was observed that the difference between the highest (about 20 MJ·$(Nm^3)^{-1}$) in the case of gasification with water) and the lowest (about 13 MJ·$(Nm^3)^{-1}$) (in the case of gasification by air) calorific value was 7 MJ·$(Nm^3)^{-1}$. The modeling results showed that the highest calorific value of the syngas was obtained at a low concentration of the gasifying agent. This is a particularly valuable outcome because one of the criteria of a gasification agent choice is the cost. In addition, it was observed that the change in temperature at a given gasification agent molar ratio does not affect the obtained calorific value of the syngas. Therefore, the gasification process can be carried out at a low temperature, which will positively affect the energy demand of the process and facilitate process control (the process does not have to be run at a specific temperature).

It was shown, that the compressed syngas may be treated as an ideal gas, taking into consideration the evaluation of a compressibility factor using the method of the weighting treatment. Especially for $H_2$, the calculated values of Z are very close to one, which can simplify further calculations of energy demand in compression for this gas. Other gasifying agents also lead to values of Z being very close to one, especially for their low molar concentrations. The determination and derivation of a $B_{syn}$ factor can aid a rapid estimation of gas volume increase in compression, which makes it possible to assess the energy content in the transport that is a direct economic result. Confirming that it is beneficial to

compress the transport and utilize the energy contained in a gasified SS may create new possibilities to generate energy and supply end-users with novel fuel in the form of compressed syngas that can be easily distributed using road, rail or water transport. The compression of syngas makes it possible to store a larger amount of gas in a given volume to increase the available energy for end-users. It was shown that compressing the syngas to 20 MPa allows the transport up to 404.61 times more gas in a given volume compared to the transport of gas at the standard pressure and temperature conditions (STP). This concept proposes a new method to utilize SS as a novel energy source that can be easily distributed in the pressure vessels to end-users using all modes of the transport capacity of carrying intermodal ISO containers.

Additionally, obtained results show that the choice of a gasification agent may not be based on the price of the agent, but on the amount of energy in the transported syngas. It was shown that the energy contained in one transport of compressed syngas is enough to meet the energy demand of an average Polish household for at least a year, which shows great potential for application and further research. Future work should include a theoretical and experimental examination of the compression and decompression process, validation of the developed numerical model and simulations and examination of the influence of the syngas composition on the pressure vessels and compressor used to compress and store it. This concept opens a completely new opportunity to utilize green fuel and should be further investigated.

**Author Contributions:** Conceptualization, M.M., P.S., and A.B.; Data curation, M.M., P.S., and A.B.; Formal analysis, M.M. and P.S.; Investigation, M.M., M.T., and P.S.; Methodology, M.M., M.T., P.S., and A.B.; Supervision, J.A.K. and A.B.; Validation, J.A.K. and A.B.; Visualization, M.M., M.T., and P.S.; Writing—original draft, M.M. and P.S.; Writing—review and editing, M.M., P.S., J.A.K., and A.B.

**Funding:** Research conducted by Marek Mysior was funded by the Ministry of Science and Higher Education in Poland, grant number 0402/0072/18, dedicated to young researchers. "The PROM Programme—International scholarship exchange of Ph.D. candidates and academic staff" is co-financed by the European Social Fund under the Knowledge Education Development Operational Programme PPI/PRO/2018/1/00004/U/001. The authors would like to thank the Fulbright Foundation for funding the project titled "Research on pollutants emission from Carbonized Refuse Derived Fuel into the environment," completed at Iowa State University. In addition, this project was partially supported by the Iowa Agriculture and Home Economics Experiment Station, Ames, Iowa. Project no. IOW05556 (Future Challenges in Animal Production Systems: Seeking Solutions through Focused Facilitation) sponsored by Hatch Act and State of Iowa funds.

**Conflicts of Interest:** The authors declare no conflict of interest.

## Abbreviations

| | |
|---|---|
| $B_{syn}$ | syngas compression ratio, (-) |
| CHP | combined heat and power; |
| CNG | compressed natural gas; |
| HHV | higher heating value, (MJ·Nm$^{-3}$); |
| ICE | internal combustion engine; |
| LHV | lower heating value, (MJ·Nm$^{-3}$); |
| LNG | liquified natural gas; |
| NG | natural gas; |
| SETS | syngas energy transport system; |
| SNG | synthetic natural gas; |
| SS | sewage sludge; |
| STP | standard temperature and pressure (273.15 K; absolute pressure of 100 kPa); |
| Z | gas compressibility coefficient, (-) |

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
