# Peer review of "Valorization of Sewage Sludge via Gasification and Transportation of Compressed Syngas"

_processes, doi:10.3390/pr7090556_

Round 1

Reviewer 1 Report

Please see my detailed comments in the attached file. 

Reviewer 2 Report

No comments. The article looks fine and significantly improved compared to the previous submission.

Author Response

Thank You very much for the review.

Round 2

Reviewer 1 Report

The authors have addressed most of the comments; they have also tried to make changes according to the reviewer's suggestions. After revisions, the quality of the manuscript has been adequately enhanced. Therefore, the manuscript could be considered for the publication in the Journal. However, there are still some editing/ syntax errors present in the manuscript which need to be corrected, hence the publishing team is advised to read the manuscript carefully before publishing.

This manuscript is a resubmission of an earlier submission. The following is a list of the peer review reports and author responses from that submission.

Round 1

Reviewer 1 Report

Outcomes from detailed simulation experiments of sludge gasification is expected to be utilized for construction of energy recovery system. However, more discussion are considered necessary about desirable characteristics of syngas and applicability of this simulation on actual planning. Detailed description of simulation and calculation, which largely affect the originality of the study, is also necessary.

Reviewer 2 Report

Line 131, provide the name of the reference and describe the study a little bit.

Please do the same for line other references. i.e., line 140, line 141

Line 144, “The process pressure was 1 atm [28].” Rearrange the sentence.

Reviewer 3 Report

Please find my detailed comments in the attached file. 

Reviewer 4 Report

The manuscript entitled "Theoretical analysis of transportation of compressed syngas from sewage sludge gasification" reports a theoretical study to examine the influence of the calorific value, developed by the syngas deriving from sewage sludge, on the efficiency of energy transport of the compressed syngas.

In this theoretical analysis several variables were considered such as the process temperature and the nature of the gasifying agents. Finally, the molar ratio of the gasifying agent / carbon contained in the sludge was considered.

Five different gasifying agents were considered such as: Air, oxygen, hydrogen, carbon dioxide, water vapor, over twenty different process temperature values and a molar ratio of the gasifying agent/carbon in the sludge of between 0.1 and 1.0.

The topic of the study is interesting and can be a valid support for researchers working in this field.

The study is detailed, precise and well treated. The introduction is complete with the presence of sub-paragraphs which make it very clear.

The method used is presented in a systematic and clear manner. The results are reported and discussed in full.

I believe that the manuscript has the characteristics to be published although during the reading minimal errors or inaccuracies emerged which are reported below:

Figure 2. H2O instead of H20. The caption and the legend should be better explained.

Table 4. H2O instead of H20

Line 245. What does  0: C mean?

Line 292 Table 5 instead of Table 6

Round 2

Reviewer 1 Report

Research articles need originally obtained data or methodologically original simulation and analyses. The reviewer consider that concluded model in revised manuscript is based on the existing data and set condition, and calculation by prepared software. Therefore, originality of this manuscript is perceived to be insufficient as a research article.

Reviewer 3 Report

Please find my detailed comments in the attached file. 
